# The effects of Urban Identity on Entrepreneurial Choice

**Yanhong Wang**[1,2]**, Haifang Feng**[1]***, Tiantian Zhang**[1]

**1** Hebei University of Economics and Business, Shijiazhuang, Hebei, China, **2** Hebei Collaborative Innovation Center for Urban-Rural Integrated Development, Shijiazhuang, Hebei, China

* 807054889@qq.com

**Data Availability Statement:** All relevant data are within the manuscript and its Supporting Information files.

**Funding:** This submission was funded by Major Program of National Fund of Philosophy and Social Science of China (19BSH046) and achievement of

## Abstract

This study investigate how the Urban identity(UI) influence the entrepreneurial choice of the migrants. Drawing on the identity economics theory in combination with microscopic perspective on entrepreneurship, we conclude that the UI increases the odds of self-employment of the migrants by 19.91% after solving endogenous problem in our sample of 126385 individuals from the China Migrants Dynamic Survey. We test the moderating effect of medical insurance and find that the interaction coefficient is positive. This study further reveals that the expanding social networks, improving urban integration, and increasing income are the three main mechanisms through which the UI influences the entrepreneurial choice of the migrants. So, we derive results consistent with our hypotheses. The findings have implications for both the entrepreneurship and national policy literature.

## 1. Introduction

Entrepreneurship is one of the main pillars of growth in any economy. Entrepreneurship has been essential in promoting economic transformation and upgrading, shifting gears, accelerating speed, and leading to new economic growth points [1]. Starting a business of the migrants in a city can not only solve their own employment problems, improve their income, and thus enhance their ability to integrate into the city, but also promote new employment growth [2]. As the core cornerstone of regional economic growth and well-being, it provides job opportunities to the place of immigration [3–5]. Achieving a high rate of entrepreneurship in region has become the priority objective of governments and firms. Therefore, a central question often debated among academics, policy makers, and ecosystem stakeholders is what can we do to spur the entrepreneurial activity of the migrants [6].

What influences entrepreneurial choice has always been a hot topic in entrepreneurship. Exploring the influence factors of entrepreneurship choice from different theoretical perspectives will help us discover the internal mechanism of the migrants' choice of entrepreneurship. Previous literature has analyzed mechanisms that improve entrepreneurship from three perspectives. The first research category starts from social cognition and considers that the key to entrepreneurship choice lies in the entrepreneur's ability and previous experience. It is investigated from the perspectives of social relations, dialect skills, parents' entrepreneurial behavior,

the 2022 Special Project Research Project of the Training and Training Center for College Ideological and Political Work Teams of the Ministry of Education (2022HYY010).

**Competing interests:** The authors have declared that no competing interests exist.

etc [3, 7–9]. The second category systematically analyzes the factors that affect the entrepreneurship choice of the migrants from the perspective of entrepreneurial environment ecology, focusing on the impact of environmental characteristics such as household registration system, urban inclusive, social security, and financial policies on entrepreneurial decision [10–12]. The third category interprets the demographic and psychological characteristics of entrepreneurs. It analyzes whether individual characteristics such as entrepreneurs' gender, education level, number of children, risk attitude, achievement demand, and so on to promote entrepreneurial choice [13, 14]. This paper attempts to shed light on the influence of psychological factors on the entrepreneurial choice of the migrants at country level. Thus, our research question is: Do social psychological factors of identity impact entrepreneurial attitudes and activity? The urban identity is the psychological factor we focus on.

Drawing on the identity economics theory in combination with microscopic perspective on entrepreneurship, we attempts to verify whether UI leads to more entrepreneurship of the migrants and to analyze the regulatory effect of the insurance between UI and entrepreneurial choice in our sample of 126385 individuals from the China Migrants Dynamic Survey. To address the research questions of this paper, we adopt the Probit and IV-Probit as our estimation method. Our results confirm the relationship of UI and the entrepreneurship of the migrants and indicate the regulatory effect of insurance. Our results are robust after controlling the measurement error, sample selection bias, and sample heterogeneity. This study further reveals the transmission mechanism of the UI on the entrepreneurial choice of the migrants from three perspectives: expanding social networks, improving urban integration, and increasing income. We find that the UI positively relate to the establishment and expansion of the migrants' social network affecting the acquisition of entrepreneurial resources and the entrepreneurial opportunities of the migrants. The UI improves willingness of the migrants to follow the social norms in the cities, reduces the discrimination of the local population, and enhances the willingness of the migrants to flow in. But simple behavioral imitation cannot promote the entrepreneurship of the migrants. Psychological integration can promote the entrepreneurship. The UI has a positive effect on income and improves the probability of entrepreneurial choice.

This study contributes to entrepreneurship literature in three main aspects. Firstly, based on the identity economics theory, we develop a theoretical framework linking entrepreneurial decision with the UI of the migrants. we highlight the positive effects of Urban Identity on the entrepreneurial choice of migrants and clarify how UI increases the opportunity of entrepreneurship. Secondly, we use extensive CMDS data, covering 169989 households in 32 provincial-level units and 360 prefecture-level cities in China. This paper also matches the CMDS data with the City Statistical Yearbook data and the Province Marketization Index data based on the city code of the inflow place. Thirdly, this study aims to Reveal mechanism of UI on the entrepreneurial choice of the migrants by articulating the path from expanding social networks, improving urban integration, and increasing income.

The remainder of the paper is structured as follows. We present the relevant literature and develop our hypotheses. In Section 3, we discuss our data, method, and review. Then, We report results and review the mechanism of influence of UI. A discussion follows, and we conclude with implications for theory and practice.

## 2. The theoretical framework and hypothesis development

The citizenization of the migrants is not only the change of their household registration status but also the psychological recognition of the place where they flow in. In recent years, economists have extensively explored the influence of social background factors, namely identity or

**Table 1. Payment matrix for $W_i$ and $A_i$.**

|  | Entrepreneurship | No feedback | Feedback |
|---|---|---|---|
| $W_i$ | $U_w$ | $U_w-\pi$ | $U-w$ |
|  | $U_a$ | $U_b$ | $U_b-c$ |

identification, on individual micro-behaviors, Akerlof and Kranton(2000) define identity as the phenomenon that individuals classify themselves to certain social categories and adopt corresponding behavioral norms. At the same time, they establish an identity economics model, assumesing that the behavior of people are subordinate to the group he/she identifies with. If the behavior contrary to the group he/she belongs to occurs, the individual will have anxiety and utility reduction. At the same time, the utility of other individuals in the same group will be damaged. Thus it cause a series of behavior games. Through identity economics model, this paper expounds on the mechanism of the influence of UI on the employment choice of the migrants.

At first, it is assumed that the migrants enter the city with homogeneity. After a period of integration, it is divided into two groups A and B, according to UI. A represents the migrant with UI contacting with civilian populations W, and B represents the migrant group without UI. $A_i \in A, B_i \in B, W_i \in W$, $A_i$ and $B_i$ respectively adopt two different employment decisions {entrepreneurship \employment}according to the code of conduct followed. The utility is $\{U_a, U_b\}$. We assume the $U_a$ and $U_b$ for the migrant according to their preferences in employment decisions taking account of wages and leisure. In view of the reputation of entrepreneurship in the city, the professional class of entrepreneurship is second only to "agency technology" [1] obviously, $U_b$ is greater than $U_a$, and W prefers the migrants for entrepreneurship.

As mentioned above, Table 1 shows the exchange payment return of the migrants $A_i$ and the urban $W_i$ with the identity of "urban people". According to the model assumption, the effect obtained by the urban person $W_i$ when the $A_i$ chooses to start a business is $U_w$. When the migrants chooses to go against its preference rules but $W_i$ has no feedback, $\pi$ indicates the psychological effect loss caused by $A_i$ on $W_i$, and the utility of $W_i$ at this time is $U_w-\pi$. If $W_i$ feeds back, W represents the cost that the compliance punishes the individual who deviates from the specification, and then the utility of $W_i$ is $U-W$. However, as the $A_i$ has the identity of urban, he often associates with urban person $W_i$. If $W_i$ gives negative feedback, it will cause the utility loss of C to the $A_i$, and the utility of the $A_i$ is $U_b-C$ at this time. The game equilibrium depends on the value of the parameters. When $\pi > w$, i.e. the negative feedback threat of $W_i$ is credible, there is a sub-game equilibrium. If $U_a > U_b-C$, $A_i$ will adopt entrepreneurial behavior; if $U_a < U_b-C$, $A_i$ will adopt employment behavior and accept negative feedback from $W_i$; When $\pi < w$, i.e. the negative feedback threat of $W_i$ is not credible, there is a unique sub-game equilibrium, $A_i$ adopts employment behavior, $W_i$ does not give feedback. the $B_i$ without identity will choose employment and obtain utility $U_b$.

Through the above simple game model, it reflects that the migrants with UI ultimately changes the individual's optimal choice of action due to the negative feedback mechanism of urban people. Although the result of the behavior depends on the setting of parameters, it also reflects the causal relationship.

The relationship between the identity and behaviour has long been a key question within social psychology [15] and has recently received attention within the entrepreneurship domain [16, 17]. An identity provides an individual with a frame of reference with which to interpret both the social situation and his/her (potential) actions [18]. As the highest level of social integration, the identity includes the integration of local society, economy and psychology [19]. The UI can effectively enhance their cultural integration and reduce regional discrimination,

and then reduce entrepreneurial barriers and transaction costs in the process of entrepreneurship [20, 21]. Further the UI promotes the reconstruction of the social networks and embeds it in the local social network [22], which is conducive to reducing information asymmetry, discovering entrepreneurial opportunities and increasing the possibility of financial lending [23]; Starting a business from a psychological point of view gives the migrants an identity symbol to show their social status [24, 25]. It is in line with the ideal career orientation of urban people.

More and more literature thinks that medical insurance or health insurance has a significant incentive effect on entrepreneurship and improves the probability of residents' entrepreneurship [26]. Families participating in medical insurance can reduce medical expenses to a certain extent, which in turn helps to enhance their risk tolerance [27]. Individuals participating in the insurance have an objective attitude towards the risk challenge, can better adapt to the new economic pattern, and are more conducive to the choice of self-employed working methods. Obviously, the coverage of medical insurance makes the migrants with UI increase the risk tendency and the probability of risky behavior. Therefore our alternative hypothesis is:

H2. Medical insurance has a positive moderating effect on entrepreneurship promotion of urban identity

According to the theory of identity economics, the floating population that values urban identity promotes social integration, increases income, and expand social network. The floating population with the urban identity shows the pursuit of an urban lifestyle, gradually converging with the urban population in value orientation and behavior habits, thus reducing the discrimination of the urban population and promoting the urban integration of the floating population. In today's increasingly developed social division of labor, occupation is increasingly becoming the main identification of identity. Occupational environment, reputation, scope, and nature of activities directly affect the direction of social mobility [28, 29], the entrepreneurial class is second only to "agency technology" [9], and entrepreneurs generally earn much more than working-class [6, 30]. Therefore, entrepreneurship of the floating population with urban identity is not only to avoid discrimination in the job market but also to use entrepreneurship as a possible way for economic development and upward mobility [31]. The migrants with the identity of urban tends to establish an "open community" through a business relationship, which promotes the entrepreneurship of the floating population by providing rich social capital including values, norms, and practices [32, 33], The source of starting capital for the migrants tends to rely on informal finance [34]. It is influenced by many factors, such as infectious diseases, gender of employees, and colonial legacy, etc [35–37]. The migrants possessing the urban social networks can raise starting capital conveniently [38].

and Therefore our last hypothesis is:

H3. urban identity can promote the floating population to expand the social network, increase urban integration and increase income, which may be the internal impact mechanism to promote the floating population entrepreneurship.

## 3. Data sources, variable descriptions and model building

### (1) Data sources

The main data source of this paper is the China Migrants Dynamic Survey (CMDS). The database was established by the National Health Commission of the People's Republic of China in 2009. It takes the annual report data of the floating population of 31 provinces and Xinjiang Construction Corps as the basic frame, The PPS method is adopted to sample the migrants over 15 years of age who have lived in the inflow area for more than one month and are not registered in the district (county, city). The survey content covers income and expenditure, employment, willingness to stay, health, social integration, and other aspects. It objectively

reflects the basic situation of the migrants and provides researchers with micro-data on the problem of Chinese migrants. the 2016 survey data is used in this paper, covering the sample number of 169989 households in 32 provincial-level units and 360 prefecture-level cities. This paper also matches the CMDS data with the 2015 City Statistical Yearbook data and the China Marketization Index data based on the city code of the inflow place. After data cleaning and matching, the number of valid samples was 126385.

## (2) Description of variables

Explained variable. The explained variable of this article is a dummy variable for whether the respondent started a business in 2016. We constructed it based on the question "What kind of employment status do you currently belong to?" in the CMDS questionnaire. The questions are constructed, and the answers of "employer" and "self-employed " are assigned "1" and the others are "0". Core explanatory variables. This study mainly analyzes the impact of identity on the entrepreneurial decision-making of the migrants. The core independent variable is the UI of the migrants. The degree of UI is measured by the subjective feelings of migrants. The expression in the questionnaire is "I feel I am already a city people". The options are divided into "totally disagree", "disagree", "basically agree" and " fully agree" according to the degree of consent. Based on the respondents' answers, the values of 4, 3, 2, and 1 respectively represent a decrease in the degree of UI. The head of household who answers "basic agreement" and "full agreement" is assigned "1" and the others are assigned "0"as required.

Control variables, there are four types of control variables in this article: (1) personal characteristics, gender, age, education, marital status, and work experience have been confirmed by more studies as important factors affecting entrepreneurship, self-employment choices are affected by his or her social network [39], according to the questionnaire "have you participated in the following local organizational activities since 2016", the social network assigns "1"—"6" according to the number of selected organizational activities. Non-agricultural household registration can significantly improve the probability of the migrants choosing self-employment. Therefore, the dummy variable of household registration nature is added to the model (non-agricultural household registration is 1, and agricultural household registration is 0). (2) Family characteristics. Family characteristics examine the family members and family assets. The variables are the floating experience of parents, the proportion of the population over 60 years of age and the population under 16 years of age in the family, family income, and whether there is urban house, whether there is rural land, and foundation. The first three variables are human capital within the family, which reflects the influence of the human capital of parents and children on entrepreneurial choice. According to the title "Have your parents ever worked or done business outside the home before you first moved/went out", construct the virtual variable of parents' mobile experience/business experience (yes = 1, no = 0), the proportion of the population over 60 years old and population under 16 years old in the family reflects the problem of family burden, which is calculated by the age of the population living together with the questionnaire; The latter three variables reflect the family wealth. Family income is Logarithmic assignment. The housing situation is constructed according to "which of the following properties does your house belong to" in the questionnaire. Answers to and "self-purchased commercial housing", "self-purchased affordable housing", "self-purchased small property housing" are assigned "1", others are assigned "0" according to "whether there is homestead and land", "Yes" are assigned "1" and "No" assigned "0" (3) Flow characteristics. The floating characteristics are controlled based on the floating years, the number of floating cities, and the floating scope. The more familiar the migrant is with the local environment due to long-term flow, the higher the probability of self-employment will be. The values are calculated

**Table 2. Descriptive statistics of major variables.**

| Variable | Mean value | Median | Standard deviation | Minimum value | Maximum |
|---|---|---|---|---|---|
| Whether to start a business | 0.38 | 0.00 | 0.49 | 0.00 | 1.00 |
| Citizen identity | 0.74 | 1.00 | 0.44 | 0.00 | 1.00 |
| registered residence | 0.22 | 0.00 | 0.41 | 0.00 | 1.00 |
| Age | 35.55 | 34.00 | 9.35 | 16.00 | 60.00 |
| Gender | 0.43 | 0.00 | 0.50 | 0.00 | 1.00 |
| Education | 10.38 | 9.00 | 3.29 | 0.00 | 19.00 |
| Marriage | 0.80 | 1.00 | 0.40 | 0.00 | 1.00 |
| Nation | 0.92 | 1.00 | 0.26 | 0.00 | 1.00 |
| Social relations | 3.11 | 3.00 | 1.19 | 1.00 | 6.00 |
| medical insurance | 0.63 | 1.00 | 0.48 | 0.00 | 1.00 |
| health condition | 0.99 | 1.00 | 0.11 | 0.00 | 1.00 |
| Number of households | 3.12 | 3.00 | 1.19 | 1.00 | 10.00 |
| Number of elderly over 60 years of age | 0.27 | 0.00 | 0.19 | 0.00 | 4.00 |
| Number of young people under 16 years of age | 0.68 | 1.00 | 0.76 | 0.00 | 5.00 |
| Working/business experience of parents | 0.37 | 0.00 | 0.73 | 0.00 | 1.00 |
| Log household income | 8.71 | 8.70 | 0.79 | 0.00 | 12.21 |
| contracted land | 0.46 | 0.00 | 0.29 | 0.00 | 1.00 |
| homestead in the hometown | 0.60 | 1.00 | 0.48 | 0.00 | 1.00 |
| urban housing | 0.25 | 0.00 | 0.43 | 0.00 | 1.00 |
| Flow time | 11.22 | 10.00 | 7.64 | 0.00 | 57.00 |
| Initial flow range | 0.55 | 1.00 | 0.50 | 0.00 | 1.00 |
| Number of mobile cities | 2.05 | 2.00 | 1.94 | 1.00 | 20.00 |

and assigned based on "When did you first leave your domicile", "how many cities did you move to in total" and "first floating scope" in the questionnaire respectively. (4) regional characteristics. Examining the employment level, urban economy, urban scale, and urban public service level of the living cities the variables are the urban unemployment rate, GDP per capita, market-oriented index of the provinces, the total population, etc. To alleviate the heteroscedasticity, GDP per capita, the total population, and fiscal expenditure per capital take the logarithm. The values of the major variables are shown in Table 2.

## (3) Model building

In order to verify the impact mechanism of urban identity on the entrepreneurship of the migrants and avoid the heteroscedasticity problem caused by the Linear Probability Model, we use Probit model. Probit model was put forward by American statistician C.R. Probit in 1958. The dependent variable of this model is binary, that is, there are only two possible outcomes: "Yes" or "No", but the independent variable can be continuous or classified. It assumes that there is a linear relationship between the probability of the dependent variable and independent variables, and then infers the degree of influence of different independent variables on the dependent variable. The following Probit model is constructed for estimation in the empirical study.

$$Pr(Y_i = 1|hp, X) = \frac{exp(\beta_0 + \beta_1 \text{Identity}_i + +\beta X_i + \varepsilon_i)}{1 + exp(exp(\beta_0 + \beta_1 \text{Identity}_i + \beta X_i + \varepsilon_i))} \tag{1}$$

(1) in the formula, the decision of the migrants to start a business($Y_i$) is a binary choice variable, if the individual i chooses to start a business, its value is 1, otherwise, it is 0. Identity is

UI of the migrants, X is the migrant characteristics and city-level control variables, and $\varepsilon$ is a random error term. The coefficients of UI ($\beta_1$) can be used to analyze Positive and negative effects on the entrepreneurial choices of the migrants. If it is significantly positive, it indicates that UI has a promoting effect on the migrants entrepreneurship.

Based on research hypothesis 2, the moderating effect of medical insurance is investigated by adding whether there is a medical insurance variable (yes = "1" and no = "0") and its interaction with the identity of urban into the Formula (2).

$$\Pr(Y_i = 1 | hp, X) = \frac{\exp(\beta_0 + \beta_1 \text{Identity}_i + \beta_2 \text{Identity}_i * BX_i + \beta_3 BX_i + \beta X_i + \varepsilon_i)}{1 + \exp(\beta_0 + \beta_1 \text{Identity}_i + \beta_2 \text{Identity}_i * BX_i + \beta_3 BX_i + \beta X_i + \varepsilon_i)} \quad (2)$$

$BX_i$ is the coefficient of $\beta_3$, it explains the impact of medical insurance on the entrepreneurial choice of the migrants. If the coefficient of the interaction item is positive, it indicates that the insurance improve the probability of starting a business.

## (4) Endogenous problems

Formulas (1) (2) examines the impact mechanism of local identity on the entrepreneurial choice of the migrants from the individual level.

Compared with workers, both entrepreneurs and self-employed people have a revenue premium [40]. It shows that starting a business improves the income of the migrants. Generally speaking, the migrants with good economic conditions would prefer to settle down in a city, obtain a local registered permanent residence and an independent residence, and enjoy the same welfare benefits as the citizens [14]. Therefore, there may be a reverse causal relationship between UI and individual entrepreneurship choice; In addition, unobservable factors, such as the natural endowments and abilities of the interviewees, may also affect the UI of interviewees and their entrepreneurial choice. That is to say, the UI is endogenous. Therefore, it is necessary to overcome the above endogenous problems by means of appropriate instrumental variables. The fair access of the migrants to public services, such as social management services and public health services, is binding in a certain extent. Similar circumstances inhibit the flow of identity of the migrants [41, 42]. The migrants admire the urban people included in community management, resulting in an independent sense of identity division [28, 43]. The convenience and availability of public services have a positive impact on the identity of the migrants [22]. The establishment of health records for the migrants in communities can provide them with continuous, comprehensive, economic and scientific public health services and basic medical services, reflecting the public services integration of the migrants into cities [44]. Obviously, the availability of public services affects the UI and residence intention of the migrants, is not related to their employment decisions. Therefore, we believe that the availability of public services is a reasonable instrumental variable. we select the proxy variable of the availability of public services based on questionnaire "whether the local community established a resident health record card for you".

The convenience and availability of public services have a positive impact on the identity of the migrants [22]. In response to this, this paper selects "whether the local residents have established a resident health file card for you" in the questionnaire as the instrumental variables of UI. The availability of public services affects the migrants "citizen identity" identification, but it has no different impact on their career choice. Therefore, we believe that the availability of public services is a reasonable instrumental variable.

## 4. Main results and robustness test

### (1) Main results

Table 3 gives the benchmark regression results of the UI on their entrepreneurial choice, in which regression (1) reports whether the UI of the whole sample has an impact on entrepreneurial choice. The results show that the regression coefficient of the UI of the migrants is 0.115. It is significant at the level of 1%. The UI can promote the entrepreneurial choice of the migrants by controlling individual and urban characteristics. The marginal effect of the urban identity of the migrants on entrepreneurial choice is exp (0.115). That is hypothesis (1) holds. The results of regression (2) show that the UI coefficient is slightly smaller than that of regression (1) and remains significant after the interaction item is added. The coefficient of the interaction item is positive and significant, indicating that the UI of the migrants with medical security has a greater marginal effect on entrepreneurial choice. As discussed above, the UI of the migrants is significantly endogenous to their entrepreneurial choice. Therefore, we used the public service availability as an instrumental variable to perform a two-stage IV Probit regression. The Wald endogenetic test in the last row of regression (3) and regression (4) shows that the UI of the migrants does have significant endogeneity, and the F value of the first-stage regression is much higher than 10, indicating that there is no weak instrumental variable problem. Regression (3) is the result of a two-stage IV Probit regression that includes only UI as an endogenous variable, using "relative housing price of a residence in the past year" as an instrumental variable. After solving the endogenous problem, the boundary effect of UI on the entrepreneurial choice of the migrants is 19.91%, slightly higher than the regression result of regression (1) (12.18%), indicating that UI can promote the probability of starting a business. Regression (4) the results of IV Probit regression in the two stages after adding the interaction item show that after controlling the endogeneity of UI, the promotion effect of UI is stable. The interaction item between UI and medical insurance is significantly positive, indicating that medical insurance amplifies the promoting effect of UI on the entrepreneurial choice of the migrants. The migrants with medical security in their places of residence will enhance the degree of urban identity to entrepreneurship.

With regard to the control variables, we find that being urban residence, being married, having parents with working or business experience and having social relations positively relate to entrepreneurial choice of the migrants. In terms of education, it is negatively associated with entrepreneurial choice of the migrants. Our results further indicate that healthy choices is not related to the entrepreneurship; in terms of characteristic aspects of the migrants, family assets, urban housing, and rural contracted land will reduce the probability of entrepreneurship. Rural residence land can promote entrepreneurship. The migrants with higher family income, larger family populations, more minors and fewer elderly people are more likely to choose entrepreneurship. For the flow characteristic variables, the flow time and the first flow range are directly proportional to the entrepreneurial probability of the migrants. Our results further indicate the number of floating cities is negatively associated with the entrepreneurial choice. The probable interpretation is that frequent mobility weakens social networks, which is an important factor in promoting entrepreneurship.

### (2) Robustness check

We tested the robustness of our results by Propensity Score Matching (PSM). As show in Table 4, after PSM with K-nearest Neighbor Matching method (k = 1, K = 3) and Kernel Matching method respectively, the matched control variables is implemented uniform distribution in the treatment group and the control group. In detail, the ATT values are 0.059,

**Table 3. Principal regression (Beta).**

| | Probit | | Two-stage IVProbit | |
|---|---|---|---|---|
| | **Basic model (1)** | **Interaction model (2)** | **Basic model (3)** | **Interaction model (4)** |
| Identity | 0.115*** | 0.107*** | 0.689*** | 0.556*** |
| | (12.49) | (11.45) | (4.51) | (3.53) |
| insurance | | 0.437*** | | 0.441*** |
| | | (45.35) | | (39.17) |
| Identity * insurance | | 0.090*** | | 1.451*** |
| | | (4.57) | | (7.03) |
| age | 0.012*** | 0.012*** | 0.010*** | 0.010*** |
| | (19.75) | (19.75) | (12.49) | (12.65) |
| gender | -0.013 | -0.013 | -0.010 | -0.007 |
| | (-1.61) | (-1.59) | (-1.24) | (-0.90) |
| education | -0.050*** | -0.050*** | -0.051*** | -0.052*** |
| | (-33.57) | (-33.58) | (-32.84) | (-32.35) |
| marriage | 0.341*** | 0.342*** | 0.335*** | 0.343*** |
| | (24.46) | (24.48) | (23.48) | (23.55) |
| nation | 0.236*** | 0.236*** | 0.225*** | 0.227*** |
| | (15.69) | (15.69) | (14.46) | (14.32) |
| hukou | 0.087*** | 0.089*** | 0.079*** | 0.113*** |
| | (7.26) | (7.40) | (6.36) | (8.34) |
| health | -0.001 | -0.001 | -0.027 | -0.026 |
| | (-0.04) | (-0.04) | (-0.80) | (-0.74) |
| social relations | 0.053*** | 0.052*** | 0.062*** | 0.055*** |
| | (13.45) | (13.35) | (13.19) | (11.20) |
| Parental experience | 0.052*** | 0.052*** | 0.060*** | 0.060*** |
| | (9.09) | (9.09) | (9.71) | (9.55) |
| Family size | 0.038*** | 0.038*** | 0.038*** | 0.038*** |
| | (7.68) | (7.67) | (7.48) | (7.35) |
| Number of elderly | -0.133*** | -0.133*** | -0.123*** | -0.118*** |
| | (-6.49) | (-6.47) | (-5.86) | (-5.53) |
| Number of minors | 0.103*** | 0.103*** | 0.106*** | 0.109*** |
| | (14.94) | (14.96) | (15.00) | (15.13) |
| family income | 0.364*** | 0.364*** | 0.362*** | 0.362*** |
| | (46.85) | (46.85) | (45.68) | (44.90) |
| Contracted land | -0.077*** | -0.078*** | -0.077*** | -0.085*** |
| | (-8.43) | (-8.48) | (-8.28) | (-8.96) |
| Homestead | 0.091*** | 0.092*** | 0.102*** | 0.110*** |
| | (9.10) | (9.15) | (9.63) | (10.16) |
| House property of the city | -0.031*** | -0.029*** | -0.101*** | -0.070*** |
| | (-3.18) | (-3.05) | (-4.79) | (-3.16) |
| Flow time | 0.015*** | 0.015*** | 0.014*** | 0.014*** |
| | (24.39) | (24.40) | (21.61) | (21.53) |
| Initial flow range | 0.024*** | 0.025*** | 0.056*** | 0.061*** |
| | (2.94) | (3.00) | (4.71) | (5.06) |
| Number of floating cities | -0.029*** | -0.029*** | -0.019*** | -0.018*** |
| | (-13.86) | (-13.81) | (-5.58) | (-5.27) |
| City characteristic variable | control | control | control | control |

(*Continued*)

**Table 3.** (Continued)

| | Probit | | Two-stage IVProbit | |
|---|---|---|---|---|
| | **Basic model (1)** | **Interaction model (2)** | **Basic model (3)** | **Interaction model (4)** |
| Constant | -0.912*** | -0.904*** | -1.730*** | -1.631*** |
| | (-7.18) | (-7.12) | (-6.84) | (-6.33) |
| Observations | 126,385 | 126,385 | 126,385 | 126,385 |
| Pseudo R2 | 0.1323 | 0.1324 | | |
| Wald test | | | 14.62*** | 66.59 *** |
| One-stage f value | | | 524.92*** | 505.63*** |

Note

* * *, * *, and * represent significant at 1%, 5%, and 10% levels respectively, with t values in brackets

0.054, and 0.074. The corresponding T values are much larger than 1.96. The mean deviation and median deviation are significantly reduced. It shows that the matching results are effective. We captured a balanced sample consisting 93358 pairs of respondents. The results of regression is provided in Table 4 with PSM, showing fairly similar effects of UI on entrepreneurial choice, as predicted by our Probit and two-stage IV Probit. Thus our empirical findings are robust.

This paper selects "I like the city where I live ", "I pay attention to the change of the city where I live " and "I am willing to be a member of the city" as three index variables representing psychological integration. We obtain the identity indicators containing the above information by the principal component analysis method. The principal component analysis screened out the first principal component by the largest variance contribution rate as the new proxy variable of UI. Table 5 presents that the UI still increases the probability of entrepreneurial choice of the migrants. Our results further indicate the less effect of new proxy variable. Considering the endogenous problem of variables (wald test), the availability of public service is still used as the instrument variable of the UI. Employing two-stage IV Probit method to check the effect between the UI and entrepreneurial choice. The findings of this study is that the new proxy

**Table 4.  UI on entrepreneurial choice of the migrants using PSM.**

| | Propensity score matching model | | |
|---|---|---|---|
| | Neighbor matching | | Kernel matching |
| | **K = 1** | **K = 3** | |
| UI | 0.211*** | 0.212*** | 0.0212*** |
| | (21.46) | (23.51) | (3.87) |
| ATT | 0.059*** | 0.054*** | 0.074*** |
| | (13.06) | (13.86) | (15.43) |
| control variables | control | control | control |
| mean deviation of primary data | 15.0 | 15.0 | 15.0 |
| mean deviation after PSM | 5.1 | 5.1 | 5.4 |
| median deviation of primary data | 11.5 | 11.5 | 11.5 |
| median deviation after PSM | 1.8 | 1.8 | 1.8 |
| Numbers after PSM | 93358 | 93358 | 93358 |

Note

* * *, * *, and * represent significant at 1%, 5%, and 10% levels respectively, with t values in brackets

**Table 5. Robustness test of new proxy variable.**

| | probit | | Two-stage probitIV | |
|---|---|---|---|---|
| | The Basic model (1) | Interaction model (2) | The Basic model (3) | Interaction model (4) |
| Psychological integration | 0.046*** | 0.044*** | 0.229*** | 0.199*** |
| | (-11.65) | (-11.14) | (4.54) | (3.87) |
| insurance | | 0.436*** | | 0.433*** |
| | | (-45.22) | | (41.81) |
| Psychological Integration * insurance | | 0.040*** | | 0.570*** |
| | | (-4.8) | | (7.26) |
| Control variables | control | control | control | control |
| Constant | -0.742*** | -0.752*** | -0.713*** | -0.527*** |
| | (-5.82) | (-5.90) | (-5.59) | (-3.23) |
| Observations | 126,385 | 126,385 | 126385 | 126385 |
| Pseudo R2 | 0.132 | 0.132 | | |
| Wald test | | | 13.41*** | 64.95*** |
| One-stage f value | | | 373.2*** | 359.48*** |

Note

* * *, * *, and * represent significant at 1%, 5%, and 10% levels respectively, with t values in brackets

variable after overcoming the endogenous is also subjective to improve the probability of the starting a new business.

Therefore, based on our findings, the causal relationship between entrepreneurial choice of the migrants and the UI is established.

## (3) Heterogeneity effect

We further distinguished the migrants by family, education, hukou and mobility range. We discusses the influence of UI on the entrepreneurial choice of the migrants under different samples. Family type refers to whether the householder has marital relationship. The married samples include first marriage, remarriage and de facto marriage. The unmarried migrants is composed of widowed and divorced samples; The migrants with College, undergraduate or graduate diploma are higher educational group, while the left are others; Household registration refers to the classification of the migrants into urban household registration and others; The scope of mobility divide the migrants into inter-provincial mobility or intra-provincial mobility.

Table 6 presents the results of sub-sample of the migrants. More precisely, Model (1) and (2) report the results of sub-samples of the different family types. We find that unmarried families are more sensitive to identity than married families. The possible reason is that unmarried families need a better economic basis to win the urban marriage competition. Model (3) and (4) reports the results of sub-samples with different education levels. Compared with other educational levels, the migrants with higher education levels is more sensitive to urban identity and is more positively associated with entrepreneurial choice, with apparent contradiction with the conclusion of the whole sample of education on entrepreneurship. Obviously, the migrants with higher education have a positive willingness to integrate into the city. In the next step, we divided the samples withe higher education into a junior college, undergraduate education, and above. Our results further indicate that the entrepreneurial probability of the migrants with a bachelor's degree or above increased by 11.5% and the entrepreneurial probability of the migrants with a bachelor's degree or above increased by 10.37%, which is not

**Table 6. Heterogeneity: Relation between the UI and entrepreneurial choice of sub-samples.**

| variable | Family type | | Type of education | | Household registration type | | Flow range | |
|---|---|---|---|---|---|---|---|---|
| | married (1) | other (2) | universities and colleges (3) | other (4) | cities and towns (5) | other (6) | trans-provincial (7) | other (8) |
| UI | 0.103*** (10.19) | 0.116*** (4.71) | 0.114*** (3.97) | 0.098*** (9.50) | 0.143*** (5.09) | 0.096*** (8.82) | 0.111*** (9.32) | 0.081*** (5.31) |
| UI*Insurance | 0.089*** (4.14) | 0.085* (1.71) | 0.046 (0.90) | 0.081*** (3.63) | 0.106** (2.19) | 0.083*** (3.38) | 0.089*** (3.53) | 0.031 (0.96) |
| insurance | 0.459*** (44.05) | 0.302*** (11.88) | 0.709*** (27.60) | 0.376*** (36.12) | 0.351*** (15.55) | 0.447*** (41.46) | 0.400*** (30.25) | 0.499*** (34.52) |
| Control variable | be | be | be | be | be | be | be | be |
| observed value | 101572 | 24813 | 23940 | 102445 | 27443 | 98942 | 69044 | 57341 |
| Pseudo R2 | 0.089 | 0.154 | 0.133 | 0.109 | 0.141 | 0.129 | 0.146 | 0.135 |

Note

* * *, * *, and * represent significant at 1%, 5%, and 10% levels respectively, with t values in brackets

statistically significant. The improvement of education level reduces the entrepreneurial probability of the migrants. The UI of migrants with bachelor's degree or above is not related to entrepreneurship. Model (5) and (6) presents the results of sub-samples of different household registration types. Compared with the migrants with rural household registration, the migrants with urban household registration is more sensitive to urban identity. The increase in urban identity improves the possibility of their economic and psychological integration, and thus improves the probability of their entrepreneurial choices. Model (7) and (8) report the results of the sub-sample of the migrants with different flow scope(inter-provincial or intra-provincial flow). The results show that the migrants with inter-provincial flow are more sensitive to identity. With increasing of UI by 10%, the probability of entrepreneurial choice of the inter-provincial migrants (11.2%) are slightly higher than that of the inter-provincial migrants(10.8%). In addition to model (3) and model (8), the interactions between UI and medical insurance were significantly positive, verifying the positive moderating effect of medical insurance on the entrepreneurship promotion of UI.

These findings support Hypothesis 1 and Hypothesis 2.

## 5. Mechanism and expansion

In this part, We analyzed three affecting approaches of the UI upon the Entrepreneurship of the migrants: expanding social network, increasing urban integration, and improving income.

### (1) Expansion of social networks

Social networks have been proven to assist the entrepreneurs in identifying opportunities, mobilizing resources [23, 45, 46] and gain legitimacy in local communities [45], as a source of venture capital, new customers, market information and psychological support [40]. Social networks positively relate to entrepreneurial choice. The UI of the migrants can enhance the social integration of the migrants, reduce the mobility of the, promote the migrants to frequently interact with those sharing a common way of life, and expand the social network. To examine the intermediary role of social networks of the migrants, we divide social networks into two dimensions, spanning social network and broad social network. The datas are Based on the questionnaire in 2016, "Who do you associate with most locally in your spare time", reflecting the spanning social network of the migrants. We assign "other locals" and "other

**Table 7. Mechanism test models.**

| variable | Probit/OLS | | IvProbit/TEM (treatment effect model) | | |
|---|---|---|---|---|---|
| | The Regression coefficient of identity | R2 | The Regression coefficient of identity | Wald | LRtestρ = 0 |
| A. social networks | | | | | |
| Spanning social network | 0.210*** (23.93) | 0.067 | 1.416*** (19.64) | 174.21*** | |
| broad Social network | 0.094*** (14.77) | 0.080 | 0.623*** (32.35) | | 562.09*** |
| B. urban integration | | | | | |
| Social norm | 0.212*** (33.71) | 0.060 | 0.935*** (29.55) | | 278.05*** |
| Social discrimination | -0.265*** (-54.33) | 0.117 | -0.885*** (-35.87) | | 274.02*** |
| Willingness to integrate | 0.584*** (89.74) | 0.056 | 1.130*** (38.91) | | 263.24*** |
| C. revenue effects | | | | | |
| income | 0.053*** (5.08) | 0.102 | 0.386*** (29.54) | | 136.88*** |

Note

\* \* \*, \* \*, and \* represent significant at 1%, 5%, and 10% levels respectively, with t values in brackets

outsiders" to "1" and others to "0";The questionnaire "Have you participated in the activities of the following organizations locally since 2016" represents broad social network. The data is assigned according to the number of organizations joined by the migrants. The larger data reflects the wider the social resources available to the migrants.

The results of effect of UI on spanning social relations and broad social relations are shown in Table 7 (part A). We applied Probit and IV Probit to investigate how UI affects spanning social relation and examine the effect of UI on spanning social relations by OLS and treatment effective model. We find that both spanning social relations and broad social relations of the migrants exhibit a positive association with the UI, regardless of the basic model or the endogenous control model. Our results indicate that UI positively affects the establishment and expansion of the social network of the migrants and further affects the access to entrepreneurial resources and the identification of entrepreneurial opportunities of the migrants.

## (2) Improve urban integration

Articles identify the following most common challenges refugee and immigrants entrepreneurs encounter: (a) seed capital, (b) language barriers, (c) location, (d) embeddedness,(e) knowledge about the local market [47, 48]. The important connotation of identity is the individual's recognition of the identity norms to which they belong and Cultural Integration [49]. It is manifested in the adaptation to the culture of the inflow area. The UI of the migrants promotes them to follow the social norms of the local people, and strengthen others' cognition of their own identity by imitating the behavior of the local people, such as similar in language use, customs, values, housing integration, etc [50, 51]. The UI can reduce the conflict between the behavior norms of the migrants and the local people, reduce the discrimination in the inflow place, narrow the psychological distance. Further, the above behaviors lead to the preference of the inner group and increase the cooperative behavior of the members in the group [52]. So the UI of the migrants can help alleviate the financial constraints and language barriers faced by the migrants in urban entrepreneurship, better embed in the local market and reduce friction and entrepreneurial barriers. As a whole, urban cultural Integration expand the expected income of entrepreneurship, and then enhance the entrepreneurial behaviors.

To examine the mediating effect of urban integration, we first selected five questions for principal component analysis based on the 2016 questionnaire: "My health habits are quite different from those of the residents", "it is more important for me to follow the customs and

Table 8. Impact analysis of urban integration on entrepreneurship (logistics model).

| variable | (1) | | (2) | | (3) | |
|---|---|---|---|---|---|---|
| | Entrepreneurial choice | Odds Ratio | Entrepreneurial choice | Odds Ratio | Entrepreneurial choice | Odds Ratio |
| Social norm | 0.002 | 1.002 | | | | |
| | (0.33) | | | | | |
| Social discrimination | | | -0.031*** | 0.970 | | |
| | | | (-3.41) | | | |
| Willingness to integrate | | | | | 0.069*** | 1.071 |
| | | | | | (10.47) | |
| Constant | -0.822*** | | -0.794*** | | -0.782*** | |
| | (-3.95) | | (-3.81) | | (-3.75) | |
| Observations | 126,385 | | 126,385 | | 126,385 | |

Note: The Odds Ratio coefficient shown in the table reflects the multiple of the change of the explained variable caused by the change of one unit of the explained variable. If Odds Ratio is greater than 1, it is a positive correlation, and if it is less than 1, it is a negative correlation

* * *, * *, and * represent significant ace 1%, 5%, and 10% levels respectively, with t values in brackets

habits of my hometown", "I like the city where I live now", "I pay attention to the changes in the city where I live now", "I am willing to integrate into the local population and become one of them". In the next step, we extracted two principal components based on the total variance, named as the willingness to follow the local social norms and the willingness to integrate psychologically respectively. We use "we feel that the locals despise me" to indicates local discrimination.

Table 7 (part B) presents the results for the effects of the UI on social norms, social discrimination and willingness to integrate. We find that the UI are positively related to urban integration. We examine the positive or negative impact of urban integration on the entrepreneurial choice of the migrants by Logistics model. As the results presented in Table 8, we find that simple behavioral imitation does not promote the entrepreneurial probability of the migrants, social discrimination is negatively associated with entrepreneurial choice, and Willingness to integrate can promote the entrepreneurial choice of the migrants.

## (3) Increase income

According to the theory and empirical research of the migrants, the motive of the migrants citizenship is mainly the improvement of income and welfare [53]. The migrants tends to move from backward areas to relatively developed areas with distinct "township-city" characteristics. After entering a higher-level city, the migrants with the UI will imitate the economic behavior of the local people. It significantly promote the employment quality and increase the income of the migrants [54], solving the liquidity constraints faced by entrepreneurship. To verify the role of the UI on income based on the 2016 questionnaire, we selects the data of question "what was your salary/net income in the last month (or the last employment)" to measure personal income. We tested the relationship between the UI and personal income by employing Mincer wage determination equation model to perform OLS and the treatment effect measurement.

Table 7 (Part C) presents the results of the UI on income. The results show that both OLS and treatment effect measurement reflects the positive promotion of identity on income. We divided samples into employees, self-employed, and employers according to the employment status to further test the heterogeneity. Following the same procedure, we divide inflow regions of the migrants into megacities, I cities, and other cities according to 2010 population census.

**Table 9. Heterogeneous impact of UI on income.**

| variable | Employment status | | | The size of the cities | | |
|---|---|---|---|---|---|---|
| | **employee** | **Self-employed** | **employer** | **Megacity** | **I city** | **Other Cities** |
| UI | 0.003 (0.42) | 0.168*** (6.33) | 0.179* (1.92) | 0.040*** (2.67) | 0.079*** (3.83) | 0.061*** (3.37) |
| Control variable | be | be | be | be | be | be |
| N | 75,696 | 41,082 | 7,328 | 34,012 | 25,926 | 66,447 |
| Pseudo R2 | 0.201 | 0.090 | 0.076 | 0.081 | 0.043 | 0.043 |

Note

* * *, * *, and * represent significant at 1%, 5%, and 10% levels respectively, with t values in brackets

Table 9 presents the results for the effects of the UI on the income of the sub-samples. We find that the UI has a positive effect on the income of self-employed people and employers, and is not statistically significant for the income of employees. Our results indicate the UI promotes the income of the migrants in all cities and the effect of I city is the most obvious.

From the above results, Hypothesis 3 is partially supported

## 6. Conclusions

Based on theoretical analysis and research hypothesis, we examine the impact of the UI on their entrepreneurial choice of the migrants by the survey data of CMDS. According with the results of our research, the causal relationship between the UI and entrepreneurial choice of the migrant is established(*Hypothesis 1*). Our results further indicate insurance work positive moderating effect on the entrepreneurial promotion of the UI(*Hypothesis 2*). Our results are robust after controlling the measurement error of urban identity, sample selection bias, and sample heterogeneity.

Further, we test the impact path of the UI on the entrepreneurial choice of the migrants from three terms, respectively expanding social networks, increasing urban integration, and raising income (*Hypothesis 3*). We find that the UI positively relate to the establishment and expansion of the migrants' social network affecting the acquisition of entrepreneurial resources and the entrepreneurial opportunities of the migrants. ON the one hand the UI improves willingness of the migrants to follow the social norms in the cities, reduces the discrimination of the local population, and enhances the willingness of the migrants to flow in. On the other hand, simple behavioral imitation cannot promote the entrepreneurship of the migrants and local discrimination reduces the entrepreneurship probability. Psychological integration can promote the entrepreneurship. The UI has a positive effect on income and improves the probability of entrepreneurial choice.

Different from other research of influencing factors on entrepreneurship, we focus on the impact mechanism of the UI on entrepreneurship choice of the migrants. we finds that urban identity psychology affects the behavior pattern of the migrants and promotes entrepreneurship. Thus, first we suggest that entrepreneurship policy should not only be economic policy, but also provide more social services. Second, our findings suggest provide a fair social environment to reduce social discrimination, stabilize the expectations of the migrants, enhance psychological integration, and improve the identity of the migrants. Third, we suggest to build network interaction platform through the enterprises, communities and other organizations to promote the establishment and expansion of social networks of the migrants. At last, Adjusting the income distribution can raise the income level of the migrants and provide better social security. We suggest the future policy should also take into account the heterogeneity of the migrants, and match the policies according to the different demands of the migrants.

Ideally, we would have used panel data to explore how the urban identity is related to the entrepreneurial choice of the migrants. Unfortunately, none of the existing panel data sets such as the Panel. The lack of panel data explains our choice to leverage cross-sectional data that based on the 2016 China Migrants Dynamic Survey. Future research should pay more attention to the dynamic influence of the UI on entrepreneurial choice. This would allow for more in-depth analyses of how the UI is related to the entrepreneurial choice of the migrants. Another clear limitation is Objective evaluation of the dependent variable. The UI in our study is measured by the subjective feelings of the individual, lacking of in-depth and comprehensive. Future research should characterize the UI from three dimensions: psychological integration, social integration and economic integration and improve the objectivity of evaluation.

## Supporting information

**S1 Data.**
(XLSX)

## Author Contributions

**Conceptualization:** Yanhong Wang.

**Data curation:** Yanhong Wang.

**Formal analysis:** Yanhong Wang.

**Funding acquisition:** Yanhong Wang.

**Investigation:** Yanhong Wang.

**Methodology:** Yanhong Wang.

**Project administration:** Yanhong Wang.

**Resources:** Yanhong Wang.

**Software:** Yanhong Wang.

**Supervision:** Yanhong Wang, Haifang Feng, Tiantian Zhang.

**Writing – original draft:** Yanhong Wang.

**Writing – review & editing:** Yanhong Wang.

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
