## [Decision Letter · Decision Letter 0]

19 Nov 2023

PONE-D-23-34376The effects of Urban Identity on Entrepreneurial Choice of Floating PopulationPLOS ONE

Dear Dr. Feng,

Thank you for submitting your manuscript to PLOS ONE. After careful consideration, we feel that it has merit but does not fully meet PLOS ONE’s publication criteria as it currently stands. Therefore, we invite you to submit a revised version of the manuscript that addresses the points raised during the review process.

We look forward to receiving your revised manuscript.

Kind regards,

Jiafu An

Academic Editor

PLOS ONE

Journal Requirements:

6. Please amend the manuscript submission data (via Edit Submission) to include author Dr. Yanhong Wang.

7. We note you have included a table to which you do not refer in the text of your manuscript. Please ensure that you refer to Table 5, 8 and 9 in your text; if accepted, production will need this reference to link the reader to the Table.

**Additional Editor Comments:**

The paper applies identity economics to study how urban identity (UI) influences the entrepreneurial choices of the floating population. Utilizing public services' availability as a proxy for UI, the study addresses endogeneity in its analysis. Analyzing data from 126,385 individuals in the China Migrants Dynamic Survey, the findings reveal that having a UI increases the likelihood of self-employment by 19.91%. The paper also examines the role of medical insurance and finds a positive correlation with UI, suggesting it amplifies the effect on entrepreneurship. The study identifies three pathways through which UI affects entrepreneurial decisions: by broadening social networks, enhancing urban integration, and boosting income. These insights contribute to the understanding of entrepreneurship and inform national policy development.

There are many recent, important papers that need to be discussed in the current study, especially in the context of China and informal institutions (i.e., urban identity). Papers that touched on Chinese economy and informal institutions should be reviewed in the currently “literature review” section. These papers include, but not limited to:

An, J., Hou, W., & Lin, C. (2022). Epidemic disease and financial development. Journal of Financial Economics, 143(1), 332-358.An, J. (2020). Is there an employee-based gender gap in informal financial markets? International evidence. Journal of Corporate Finance, 65, 101737.Guo, S., & An, J. (2022). Does terrorism make people pessimistic? Evidence from a natural experiment. Journal of Development Economics, 155, 102817.An, J., Lin, C., & Tai, M. (2023). Colonial Legacy and Informal Finance. Available at SSRN 3862205.An, J., Hou, W., & Zhang, Y. (2019). China’s rule of law in New Era: the rise of regulation and formalism. Journal of Chinese Economic and Business Studies, 17(3), 313-318.An, J., Armitage, S., Hou, W., & Liu, X. (2020). Do checks on bureaucrats improve firm value? Evidence from a natural experiment. Accounting & Finance, 60(5), 4821-4844.

Currently, the paper is not professionally organized. Figures and tables are not structured to the academic publication standard. I suggest the authors to follow the paper titled “Initial coin offerings and entrepreneurial finance: the role of founders’ characteristics.” Published in the Journal of Alternative Investment, by An, Jiafu; Duan, Tinghua, Hou, Wenxuan. In particular, every table should have a self-contained description.

I also strongly recommend the authors to consider adding a “hypothesis development” section to set the theoretical stage of why the authors design such tests. Xu, J. et al. (2023) (Inherited trust and informal finance. Journal of Business Finance & Accounting) offer a good example.

Reviewers' comments:

Reviewer's Responses to Questions

**Comments to the Author**

1. Is the manuscript technically sound, and do the data support the conclusions?

Reviewer #1: Partly

2. Has the statistical analysis been performed appropriately and rigorously? 

Reviewer #1: Yes

3. Have the authors made all data underlying the findings in their manuscript fully available?

Reviewer #1: No

4. Is the manuscript presented in an intelligible fashion and written in standard English?

Reviewer #1: Yes

5. Review Comments to the Author

Reviewer #1: 1. The reasons why urban identity is selected as the psychological factor concerned seem to be quite brief in your paper. It would be helpful if you could provide more evidence of the importance of urban identity on Entrepreneurial Choice of Floating Population.

2. In the model building section, you mentioned using the Probit model for estimation. It is recommended to provide some background knowledge about the Probit model before introducing it to help readers understand why you chose this model.

3. When addressing endogeneity issues, you chose the availability of public services as an instrumental variable. It is advisable to provide more theoretical basis or other supporting evidence when introducing instrumental variables to explain why you chose this variable.

4. When discussing the mediating effect of urban integration, it is recommended to provide some explanations of why urban integration would affect entrepreneurial decisions. You mentioned that urban integration can reduce the conflict between the behavioral norms of the floating population and the receiving city, reduce discrimination against the floating population in the receiving city, and narrow the psychological distance between the floating population and the receiving city. Please further explain why these factors would affect entrepreneurial decisions and provide relevant theoretical basis or research support.

5. It is recommended to provide more discussion of the limitations of the study and future research directions. This will help readers better understand the scope of the research and potential research extensions.

6. PLOS authors have the option to publish the peer review history of their article (what does this mean?). If published, this will include your full peer review and any attached files.

Reviewer #1: No

---

## [Author Response · Author response to Decision Letter 0]

13 Dec 2023

1：The relationship between the identity and behaviour has long been a key question within social psychology 1 and has recently received attention within the entrepreneurship domain 2-3. An identity provides an individual with a frame of reference with which to interpret both the social situation and his/her (potential) actions 4.As the highest level of social integration,the identity includes the integration of local society, economy and psychology5. The UI can effectively enhance their cultural integration and reduce regional discrimination,and then reduce entrepreneurial barriers and transaction costs in the process of entrepreneurship6-7.Further the UI promotes the reconstruction of the social networks and embeds it in the local social network8, which is conducive to reducing information asymmetry, discovering entrepreneurial opportunities and increasing the possibility of financial lending9; Starting a business from a psychological point of view gives the migrants an identity symbol to show their social status10-11.It is in line with the ideal career orientation of urban people.:

2：Probit model was put forward by American statistician C.R. Probit in 1958. The dependent variable of this model is binary, that is, there are only two possible outcomes: "Yes" or "No", but the independent variable can be continuous or classified. It assumes that there is a linear relationship between the probability of the dependent variable and independent variables, and then infers the degree of influence of different independent variables on the dependent variable.

3：The fair access of the migrants to public services, such as social management services and public health services, is binding in a certain extent. Similar circumstances inhibit the flow of identity of the migrants1-2. The migrants admire the urban people included in community management,resulting in an independent sense of identity division3-4.The convenience and availability of public services have a positive impact on the identity of the migrants5 .The establishment of health records for the migrants in communities can provide them with continuous, comprehensive, economic and scientific public health services and basic medical services, reflecting the public services integration of the migrants into cities6.Obviously, the availability of public services affects the UI and residence intention of the migrants, is not related to their employment decisions.Therefore, we believe that the availability of public services is a reasonable instrumental variable.we select the proxy variable of the availability of public services based on questionnaire "whether the local community established a resident health record card for you".

4：Promote urban Cultural Integration 

Articles identify the following most common challenges refugee and immigrants entrepreneurs encounter: (a) seed capital, (b) language barriers, (c) location, (d) embeddedness,(e) knowledge about the local market . The important connotation of identity is the individual's recognition of the identity norms to which they belong and Cultural Integration. It is manifested in the adaptation to the culture of the inflow area.The UI of the migrants promotes them to follow the social norms of the local people,and strengthen others' cognition of their own identity by imitating the behavior of the local people,such as similar in language use, customs, values, housing integration, etc.The UI can reduce the conflict between the behavior norms of the migrants and the local people, reduce the discrimination in the inflow place, narrow the psychological distance .Further, the above behaviors lead to the preference of the inner group and increase the cooperative behavior of the members in the group.So the UI of the migrants can help alleviate the financial constraints and language barriers faced by the migrants in urban entrepreneurship, better embed in the local market and reduce friction and entrepreneurial barriers . As a whole, urban cultural Integration expand the expected income of entrepreneurship, and then enhance the entrepreneurial behaviors .

5：Ideally, we would have used panel data to explore how the urban identity is related to the entrepreneurial choice of the migrants . Unfortunately, none of the existing panel data sets such as the Panel. Study of Entrepreneurial Dynamics I and II can be used as they only provide information for single countries and a substantially smaller number of individuals. The lack of panel data explains our choice to leverage cross-sectional data that based on the 2016 China Migrants Dynamic Survey.Future research should pay more attention to the dynamic influence of the UI on entrepreneurial choice.This would allow for more in-depth analyses of how the UI is related to the entrepreneurial choice of the migrants.Another clear limitation is Objective evaluation of the dependent variable .The UI in our study is measured by the subjective feelings of the individual,lacking of in-depth and comprehensive. Future research should characterize the UI from three dimensions:psychological integration, social integration and economic integration and improve the objectivity of evaluation.

---

## [Editor Report · Decision Letter 1]

18 Dec 2023

The effects of Urban Identity on Entrepreneurial Choice of the migrants

PONE-D-23-34376R1

Dear Dr. Feng,

We’re pleased to inform you that your manuscript has been judged scientifically suitable for publication and will be formally accepted for publication once it meets all outstanding technical requirements.

Kind regards,

Jiafu An

Academic Editor

PLOS ONE

Additional Editor Comments (optional):

The authors have addressed all the reviewer's detailed comments. Therefore I am happy to accept this paper.
---

## [Editor Report · Acceptance letter]

2 Jan 2024

PONE-D-23-34376R1 

PLOS ONE

Dear Dr. Feng, 

I'm pleased to inform you that your manuscript has been deemed suitable for publication in PLOS ONE. Congratulations! Your manuscript is now being handed over to our production team.

Kind regards, 

on behalf of

Dr. Jiafu An 

Academic Editor

PLOS ONE